# Two-Dimensional Full-Waveform Joint Inversion of Surface Waves Using Phases and Z/H Ratios

**Chao Zhang** [1,2,*], **Ting Lei** [2] **and Yi Wang** [3,*]

1   College of Oceanography, Hohai University, Nanjing 210098, China
2   Department of Earth Sciences, University of Toronto, Toronto, ON M5S 3B1, Canada; leiting@mail.utoronto.cn
3   State Key Laboratory of Geodesy and Earth's Dynamics, Innovation Academy for Precision Measurement Science and Technology, Chinese Academy of Sciences, Wuhan 430077, China
*   Correspondence: czhang18@hhu.edu.cn (C.Z.); ghostzzz@mail.ustc.edu.cn (Y.W.);
    Tel.: +86-155-5546-2977 (C.Z.); +86-180-0842-4426 (Y.W.)

**Abstract:** Surface-wave dispersion and the Z/H ratio are important parameters used to resolve the Earth's structure, especially for S-wave velocity. Several previous studies have explored using joint inversion of these two datasets. However, all of these studies used a 1-D depth-sensitivity kernel, which lacks precision when the structure is laterally heterogeneous. Adjoint tomography (i.e., full-waveform inversion) is a state-of-the-art imaging method with a high resolution. It can obtain better-resolved lithospheric structures beyond the resolving ability of traditional ray-based travel-time tomography. In this study, we present a systematic investigation of the 2D sensitivities of the surface wave phase and Z/H ratio using the adjoint-state method. The forward-modeling experiments indicated that the 2D phase and Z/H ratio had different sensitivities to the S-wave velocity. Thus, a full-waveform joint-inversion scheme of surface waves with phases and a Z/H ratio was proposed to take advantage of their complementary sensitivities to the Earth's structure. Both applications to synthetic data sets in large- and small-scale inversions demonstrated the advantage of the joint inversion over the individual inversions, allowing for the creation of a more unified S-wave velocity model. The proposed joint-inversion scheme offers a computationally efficient and inexpensive alternative to imaging fine-scale shallow structures beneath a 2D seismic array.

**Keywords:** adjoint tomography; Rayleigh wave phase; Z/H ratio; joint inversion; 2D seismic array

## 1. Introduction

Surface waves are essential for resolving crust and upper mantle structures at various length scales. Surface waves have dispersive characteristics; that is, a surface wave travels with different velocities at different frequencies in the wave-propagation process. This feature has been well studied and widely used to image Earth structures, especially for the S-wave velocity model. Teleseismic surface-wave tomography was used in earlier studies, and it offered a good resolution for the upper mantle [1–3]. However, it is difficult to constrain structures in the shallow to middle crust due to the low frequency of such waves. In recent decades, ambient-noise tomography was developed, which can extract the Green's function of a surface wave from ambient-noise data, and it significantly improved the surface-wave-dispersion data coverage (mostly less than a 40 s period). Short- to intermediate-period surface waves have been widely used for imaging regional- or local-scale crustal structures [4–7].

However, the surface-wave-dispersion curve is only sensitive to the integral feature of the velocity structure of a depth range, resulting in ambiguities during inversion. Studies have shown that the amplitude information of a surface wave can also help improve the velocity structure. The Z/H ratio (or ellipticity) of a Rayleigh wave is always defined as the amplitude ratio of the vertical and horizontal components of the fundamental mode of the Rayleigh wave. The Z/H ratio of a Rayleigh wave is only controlled by the structure

beneath the recording seismic station [8,9], and it is more sensitive to shallow structures than dispersion in the same period range. Therefore, the Rayleigh wave's Z/H ratio can provide extra constraints to reduce the ambiguity of surface wave tomography, and in particular, it can improve the imaging of shallow sedimentary structures. Because of the resolution gap caused by the different data sensitivities, independent inversions using the surface-wave-dispersion curve or the Z/H ratio cannot provide a unified S-wave velocity model. The joint inversion of the surface-wave-dispersion curve and the Z/H ratio has been proposed for imaging the complementary velocity structure. In particular, joint inversion based on ray theory has been widely used in imaging crust and upper mantle structures of various length scales [10]. However, this approach does not account for the complexities of wave propagation in complex media, and may lead to inaccurate tomographic results.

Recent advances in the numerical methods involving the wave equation combined with developments in high-performance computation (HPC) have enabled the routine simulation of seismic wave propagation in realistic 2D and 3D Earth models based on the spectral-element method (SEM) (Komatitsch and Tromp, 2002a, 2002b). Utilizing these seismic-wave simulations, adjoint-state methods can efficiently incorporate the full nonlinearity of the wave propagation into iterative seismic inversions [11–22]. Based on these advances, Chen et al. [23] proposed ambient-noise adjoint tomography for 3D crustal structures by matching the empirical Green's function (EGF) and the synthetic Green's function (SGF) calculated based on the SEM. In addition, full-wave 3D ambient-noise tomography has been developed, and has been used to image the structure of the Cascades, the Ontong Java Plateau, and the eastern North American margin [24–26]. Bao et al. [27] derived the equations for 3D full-wave sensitivity kernels of the Z/H ratio of a Rayleigh wave using the scattering-integral method [28–31].

However, the computational cost and storage requirement of 3D adjoint tomography are large. In contrast, 2D adjoint tomography is much more computationally efficient. Currently, an increasing number of dense linear arrays have been deployed around the world, which facilitates the high-resolution imaging of the 2D structures beneath the linear array. In global-scale tomography, Zhang et al. [32] presented a 2D ambient-noise adjoint tomography technique for a linear array with a significant reduction in computational cost, and applied it to an array in northern China. Tong et al. [33] developed 2D teleseismic body-wave adjoint tomography by an FK-SEM hybrid method. Zhang et al. [34] proposed a strategy of joint inversion for ambient-noise surface waves and teleseismic body waves recorded by linear arrays based on the adjoint-state method, which can be used to yield a fine S-wave velocity model and Moho topography. Small-scale tomography has been fully developed by the oil industry in the past three decades (well known as FWI), while 2D elastic FWI has recently been applied to shallow surface waves [35–38]. In particular, Forbriger et al. [39], Schäfer et al. [40], and Groos et al. [41] proposed a stable workflow for the 2D FWI of shallow-seismic Rayleigh waves not only for synthetic data, but also for real field-recorded data. However, including surface waves in FWI is a challenging issue, especially the cycle-skipping problems that occur due to these waves' higher amplitudes and different physical behaviors with respect to body waves. Benefiting from the fact that the sensitivity to the S-wave velocity is only concentrated near stations, the Z/H ratio information can constrain the shallow S-wave velocity structure in the near surface, which helps mitigate the cycle-skipping problem.

However, there is a lack of research about FWI using the Z/H ratio or the joint inversion of both data sets. In this paper, we present a systematic investigation of the 2D Rayleigh wave travel-time (phase) sensitivity kernel and the Z/H-ratio kernel via an adjoint-state method [19]. In addition, a full-waveform joint-inversion scheme for surface waves with phases and Z/H ratios is proposed to take advantage of their complementary sensitivities to the S-wave velocity structure. Finally, synthetic-imaging experiments are described to demonstrate the potential of the proposed method to yield a better complementary S-wave velocity model with a high-resolution.

## 2. Calculation of 2D Rayleigh Wave Phase and Z/H-Ratio Kernel

Tanimoto and Rivera [9] calculated the 1D depth-sensitivity kernel of the Rayleigh wave dispersion and the Z/H ratio either theoretically or numerically by perturbing the model parameters at various depths. The features of the 1D kernel include that both the dispersion and Z/H ratio are sensitive to changes in the S-wave velocity and the density, but they are less sensitive to the P-wave velocity. For the dispersion, the S-wave velocity sensitivity is positive at different depths, suggesting that the dispersion provides more constraints on the integral velocity feature. In contrast, the S-wave velocity sensitivity of the Z/H ratio changes from positive near a free surface to negative at deeper depths, implying that the Z/H ratio resolves the velocity contrasts better. To study the features of the 2D kernel, we derived equations for the 2D full-wave sensitivity kernels of the phases and the Z/H ratio based on the adjoint-state theory.

### 2.1. D Full-Wave Sensitivity Kernels of the Traveltime Equation

First, we briefly review the travel-time and amplitude-sensitivity kernels for single-component seismograms. Following the full-wave finite-frequency theory, the perturbations in the overall misfit $\delta\chi$ are linearly related to the relative perturbations in the density ($\delta ln\,\rho$), P-wave velocity ($\delta ln\,\alpha$), and S-wave velocity ($\delta ln\,\beta$) :

$$\delta\chi = \int \left[ K_\alpha(\boldsymbol{r})\delta ln\alpha + K_\beta(\boldsymbol{r})\delta ln\beta + K_\rho(\boldsymbol{r})\delta ln\rho \right] d^2x, \tag{1}$$

In general, two primary measurements are used in adjoint tomography: the phase delay and the amplitude reduction. For the phase-delay measurements, the travel-time misfit perturbations ($\delta t$) can be represented as follows:

$$\delta t = -\frac{\int_{t_1}^{t_2} \partial\widetilde{s}(t)\delta s(t)dt}{\int_{t_1}^{t_2} |\widetilde{s}(t)|^2 dt} = \int \left[ K_\alpha^p(\boldsymbol{r})\delta ln\alpha + K_\beta^p(\boldsymbol{r})\delta ln\beta + K_\rho^p(\boldsymbol{r})ln\rho \right] d^2x, \tag{2}$$

where $\delta t$ is the travel-time anomaly; $\delta s(t)$ is the perturbation of the waveform displacement, $\widetilde{s}(t) - s(t)$; the tilde represents the synthetics based on the reference model; and $t_1$ and $t_2$ are the minimum and maximum values of the time window, respectively. $\partial$ is the time derivative, so $\delta t$ is equal to the lag time at the maximum of the cross-correlogram between the observed and synthetic waveform. $K_\alpha^p$, $K_\beta^p$, and $K_\rho^p$ are the travel-time Fréchet kernels for the density, P-wave velocity, and S-wave velocity, respectively, which can be calculated using the adjoint-state method. The adjoint source of the cross-correlation travel-time is:

$$f_p^\dagger(t,\omega) = -\frac{w(t,\omega)\dot{s}(t,\omega)}{\int w(t,\omega)\ddot{s}^2(t,\omega)dt}, \tag{3}$$

where $w(t,\omega)$ is the time window over which the measurement of the cross-correlation traveltime is made, $s\,(t,\omega)$ is the waveform recorded by a station, the dot represents the time derivative, and $\omega$ is the central frequency of the Gaussian filter.

### 2.2. D Full-Wave Sensitivity Kernels of the Z/H Ratio Equation

For the amplitude reduction measurements, the natural logarithm of the amplitude reduction ($\delta$ln $A$) can be expressed as follows:

$$\delta \ln A = -\frac{\int_{t_1}^{t_2} \widetilde{s}(t)\delta u(t)dt}{\int_{t_1}^{t_2} |\widetilde{s}(t)|^2 dt} = \int \left[ K_\alpha^q(\boldsymbol{r})\delta ln\alpha + K_\beta^q(\boldsymbol{r})\delta \ln\beta + K_\rho^q(\boldsymbol{r})\delta ln\rho \right] d^2x, \tag{4}$$

Here, A is defined as the ratio between (1) the amplitude of the cross-correlogram of the observed and synthetic waveforms at lag time $\delta t$ and (2) the maximum amplitude of the auto-correlogram of the synthetic waveform. $K_\alpha^q$, $K_\beta^q$, and $K_\rho^q$ are the corresponding

amplitude-reduction Fréchet kernels for the density, P-wave velocity, and S-wave velocity, respectively. The Z/H ratio (η) of a Rayleigh wave is always defined as the Z-component's amplitude ($A_Z$) over the H-component's amplitude ($A_H$) at a certain frequency $\omega$:

$$\eta(\omega) = \frac{A_Z(\omega)}{A_H(\omega)}, \tag{5}$$

The perturbation of the Z/H ratio can be expressed as the difference between the Z-component and H-component of the amplitude perturbation:

$$\delta \ln \eta = \delta\left(\ln A^Z / A^H\right) = \delta \ln A^Z - \delta \ln A^H, \tag{6}$$

If we consider the fact that the variation in the measurements is only caused by the variation in the S-wave velocity ($\beta$), then

$$\delta \ln \eta = \int K_\beta^\eta \delta \ln \beta d^2 x; \ \delta \ln A_Z = \int K_\beta^Z \delta \ln \beta d^2 x; \ \delta \ln A_H = \int K_\beta^H \delta \ln \beta d^2 x, \tag{7}$$

where $K_\beta^\eta$, $K_\beta^Z$, and $K_\beta^H$ are the sensitivity kernels to the S-wave velocity of the Z/H ratio, the Z-component's amplitude, and the H-component's amplitude, respectively.

$$\int K_\beta^\eta \delta \ln \beta d^2 x = \int K_\beta^Z \delta \ln \beta d^2 x - \int K_\beta^H \delta \ln \beta d^2 x, \tag{8}$$

This equation holds true for any volume V, so the Z/H-ratio kernel of the S-wave velocity can be expressed as:

$$K_\beta^\eta = K_\beta^Z - K_\beta^H, \tag{9}$$

Similarly, we can obtain the following equations for P-wave velocity ($\alpha$) and density ($\rho$):

$$K_\alpha^\eta = K_\alpha^Z - K_\alpha^H, K_\rho^\eta = K_\rho^Z - K_\rho^H, \tag{10}$$

Thus, to generate the kernels above, we only need to calculate the sensitivity kernels of the Z-component's amplitude and the H-component's amplitude, which can also be solved by using the adjoint-state method. The adjoint source of the amplitude is:

$$f_q^\dagger(t, \omega) = \frac{w(t, \omega)s(t, \omega)}{\int w(t, \omega)s^2(t, \omega)dt}, \tag{11}$$

where $w(t, \omega)$ is the time window over which the measurement of the amplitude is made, $s(t, \omega)$ is the waveform recorded by a station, and $\omega$ is the central frequency of the Gaussian filter. The definition of Z/H ratio in this study is noteworthy. Generally, the previous studies [7,27] commonly used the maximum amplitude of the envelope to calculate Z/H ratio as follows:

$$\eta(\omega) = \frac{A_z(\omega)}{A_H(\omega)} = \frac{\max(E[s_z(\omega)])}{\max(E[s_H(\omega)])}, \tag{12}$$

where $s_z(\omega)$ and $s_H(\omega)$ are the Z-component and H-component of the seismogram processed using a Gaussian filter at a certain frequency $\omega$, respectively, and $E$ is their envelope. In this study, we used the following equation defined in [42]:

$$\eta(\omega) = \frac{A_z(\omega)}{A_H(\omega)} = \sqrt{\frac{\int_{t_1}^{t_2} s_Z^2 dt}{\int_{t_1}^{t_2} s_H^2 dt}}, \tag{13}$$

where $t_1$ and $t_2$ are the start and end of the measurement window, respectively. With respect to physical properties, both definitions in Equations (12) and (13) are robust and acceptable, and could result in very similar Z/H ratios for synthetic tests. In the real

data application, they will be different, depending on the S/N level of the waveform data. Determining which is better would require a more detailed discussion regarding a different real data set. Otherwise, it should be noted that we need to guarantee that the same definition is used in the data processing and kernel calculation.

### 2.3. Sensitivity Calculation and Analysis

To illustrate the procedure for obtaining the travel-time and Z/H-ratio-sensitivity kernels, we used the 2D spectral-element method (SPECFEM2D) to calculate the 2D sensitivity kernels via the adjoint-state method. The numerical parameters of the modeling in the computation are shown in Table 1. The model measures 800 km in the horizontal direction, with 320 uniform mesh nodes (quadrangles in 2D), and 100 km in the vertical direction, with 40 uniform nodes. A total of $(4 \times 320 + 1) * (4 \times 40 + 1) = 206{,}241$ unique grid points were used for the 2D spectral-element discretization with a four-degree polynomial (Komatitsch et al., 2005). The average grid spacing of the mesh was 2.5 km, which was sufficient to accurately simulate the surface waves at periods of greater than ~8 s (Komatitsch and Tromp, 2002). For simplicity and to minimize the possible interference of other arrivals on the Rayleigh waves, the P-wave velocity, the S-wave velocity, and the density were all held constant (Vp = 6000 m/s, vs. = 3500 m/s, and $\rho$ = 2800 g/m$^3$). The time interval of the simulation was set to 0.02 s to satisfy the numerical stability, and the time step was set to 12,000. We placed a Gaussian-type source-time function with a dominant frequency of 0.2 Hz at location (650 km, 0 km) to generate the 2D synthetic waveform of the surface wave, which was computed using SPECFEM2D. A station was set at location (150 km, 0 km).

**Table 1.** Numerical parameters for kernel calculation in the computation.

| Model Size (Nx × Nz) | x-Direction: 800 km; z-Direction: 100 km (300 × 40 Elements) |
|---|---|
| Material properties | Vp = 6000 m/s, vs. = 3500 m/s, $\rho$ = 2800 g/m$^3$ |
| Propagation time | 12,000 time steps; Time discretization: 0.02 s |
| Type of source-time function | Gaussian |
| Dominant frequency of source | 2 Hz |
| Source location | (650 km, 0 km) |
| Station location | (150 km, 0 km) |

Figure 1 illustrates how to obtain the single-component phase sensitivity of the kernels $(K_\beta^{pZ}, K_\beta^{pH})$ to $\delta Vs$ on a vertical profile and horizontal profile in 2D media for surface waves with a period of 20 s. First, we obtained the seismograms of the Z-component (Figure 1a) and H-component (Figure 1b) recorded by the station, which were processed using a Gaussian filter with a central frequency of 0.5 Hz. Then, we obtained the adjoint source of the Z-component (Figure 1c) and H-component (Figure 1d) at the station using Equation (3). We performed one adjoint simulation by injecting the back adjoint source at the station into the model, which created adjoint wavefield $s^\dagger$. Finally, the phase-sensitivity kernels of the Z-component (Figure 1e) and H-component (Figure 1f) at periods of 20 s were obtained using the time integration of the interaction of $s^\dagger$ and the forward wavefield $s$. To study the features of the phase kernel in the depth and lateral directions at different periods, the Z-component kernels at three other periods of 10 s (Figure 1g), 30 s (Figure 1i), and 40 s (Figure 1j) also were calculated using the same procedure.

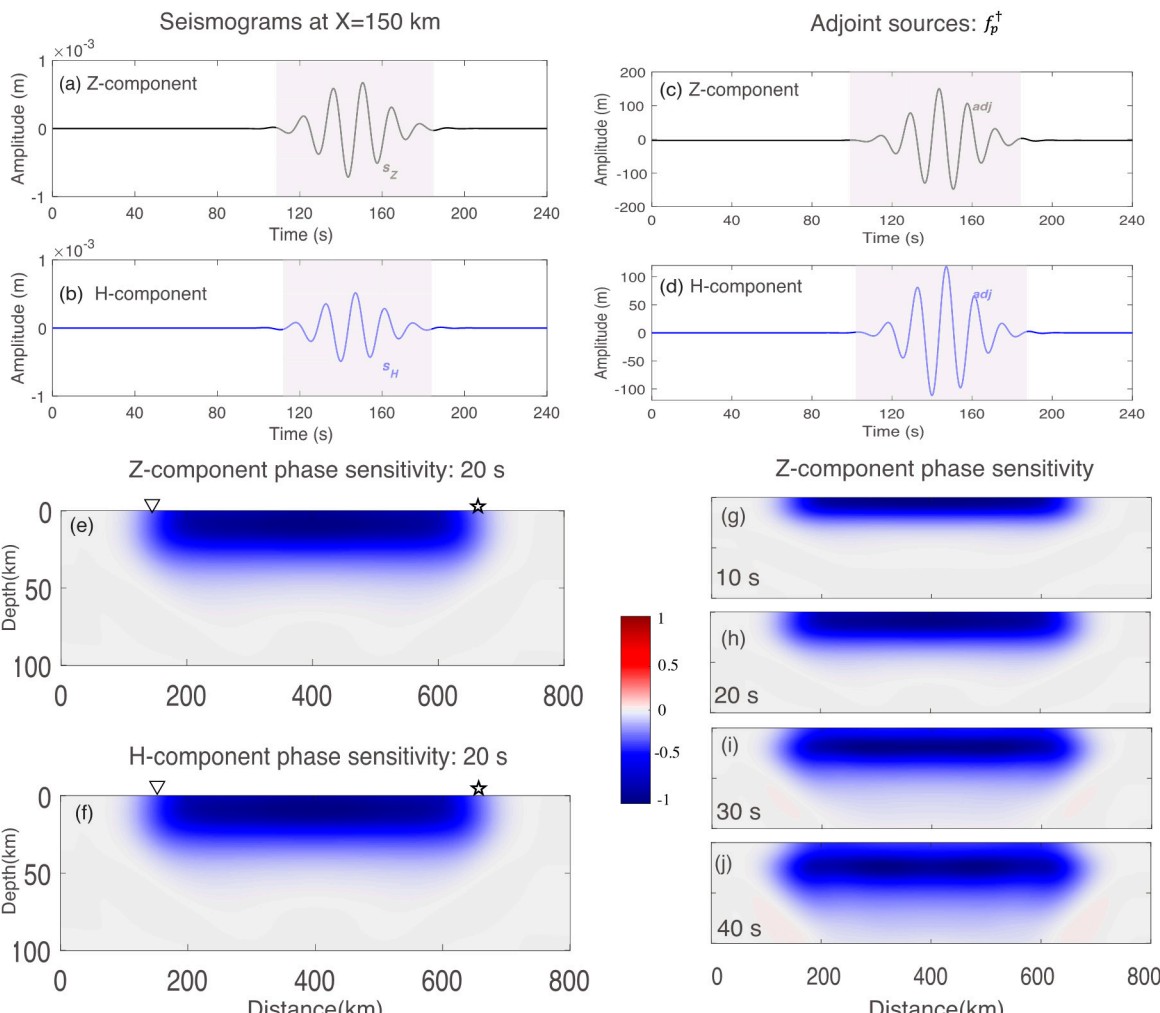

**Figure 1.** Procedure for obtaining the phase-sensitivity kernels for the surface waves with a period of 20 s, for a source (star) and station (triangle) pair. Seismograms were filtered using a Gaussian filter (central frequency: 0.5 Hz) recorded by the station for the (**a**) Z-component, (**b**) H-component, (**c**) adjoint source of the Z-component, and (**d**) adjoint source of the H-component. The shaded area represents the time window over which the measurement of the phase delay was made. (**e**) The phase-sensitivity kernels of the Z-component, and (**f**) the phase-sensitivity kernels of the H-component. The Z-component phase-sensitivity kernels at other periods of (**g**) 10 s, (**h**) 20 s, (**i**) 30 s, and (**j**) 40 s were also calculated to study the features of the phase kernel in the depth and lateral directions at different periods.

Figure 2 illustrates how to obtain the single-component amplitude kernels ($K_\beta^{pZ}$, $K_\beta^{pH}$) to δVs on a vertical profile and horizontal profile in 2D media, as well as their differential kernels ($K_\beta^{pZ} - K_\beta^{pH}$), which is called the Z/H-ratio kernel ($K_\beta^{\eta}$) in this paper. The procedure for obtaining the amplitude kernels was very similar to that for the phase kernels, except that the adjoint source was generated using different measurements followed by Equation (11). The amplitude-sensitivity kernels of the Z-component and H-component with periods of 20 s obtained using the adjoint-state method are shown in Figure 2e,f. Thus, the Z/H-ratio-sensitivity kernel at a period of 20 s (Figure 2g) could be obtained by subtracting the H-component amplitude-sensitivity kernel from the Z-component amplitude-sensitivity kernel. Similarity, the Z/H-ratio kernel at periods of 10 s (Figure 2h), 30 s (Figure 2j), and 40 s (Figure 2k) are also displayed and will be discussed later in this section.

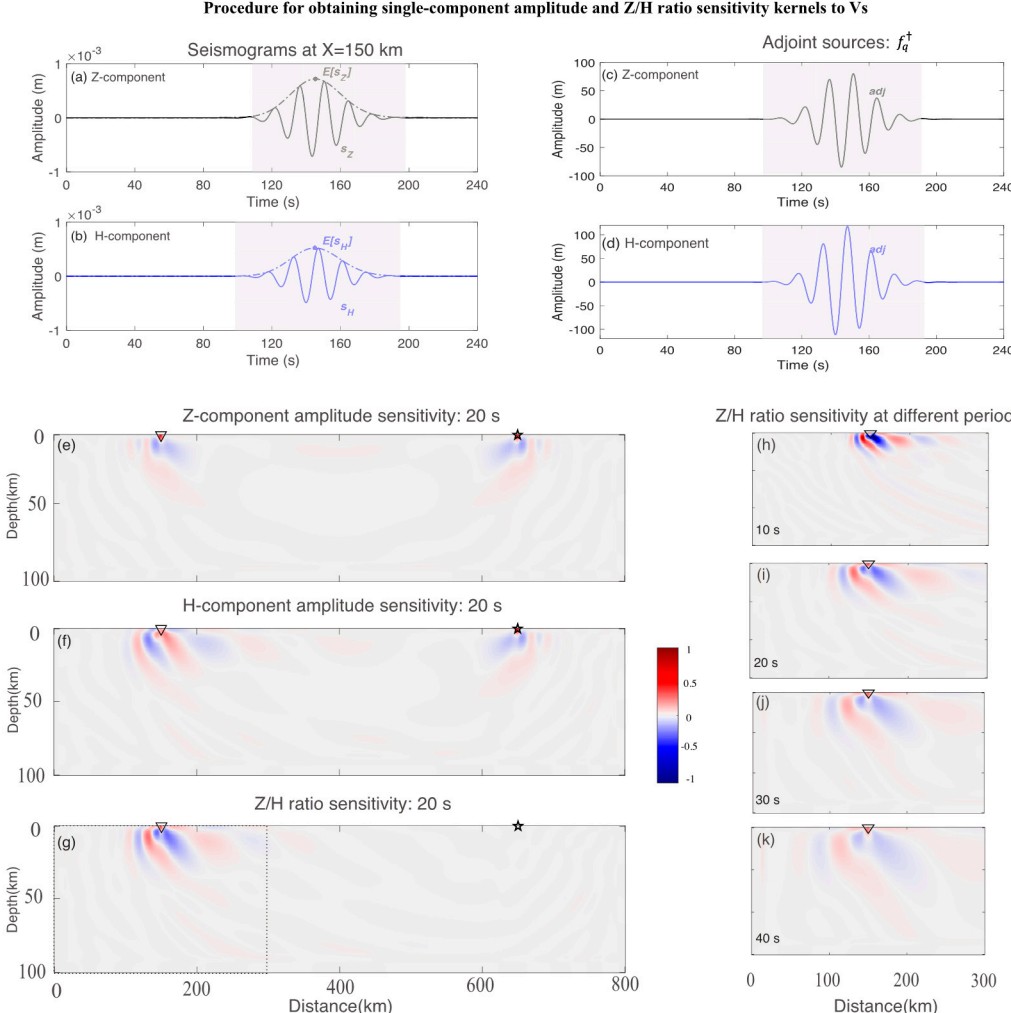

**Figure 2.** Procedure for obtaining the amplitude- and Z/H-ratio-sensitivity kernels for surface waves with a period of 20 s, for a source (star) and station (triangle) pair. Seismograms were filtered using a Gaussian filter (central frequency: 0.5 Hz) recorded by the station for the (**a**) Z-component, (**b**) H-component, (**c**) adjoint source of the Z-component, and (**d**) adjoint source of the H-component. The shaded area represents the time window over which the measurement of the amplitude reduction was made. (**e**) The amplitude-sensitivity kernels of the Z-component, and (**f**) the amplitude-sensitivity kernels of the H-component. (**g**) The Z/H-ratio-sensitivity kernels obtained by subtracting the H-component amplitude-sensitivity kernel from that of the Z-component. The Z-component Z/H-ratio kernels at other periods of (**h**) 10 s, (**i**) 20 s, (**j**) 30 s, and (**k**) 40 s were also calculated to study the features of the Z/H kernel in the depth and lateral directions at different periods. Note that only the kernels near the station (the region outlined by the dashed line in (**g**)) were plotted.

The experiment described above shows that both the 2D-phase and Z/H-ratio kernels provide more information than the 1D sensitivity kernels. The 1D kernels only had sensitivities in the vertical direction because they did not consider the source direction. In contrast, the 2D kernels varied with distance from the station and with depth, which indicated that they had strong resolving powers in both the depth and lateral directions.

Both kernels had different features in the vertical and lateral directions. The phase kernel was concentrated along the propagation path, with a much broader sensitivity range than the Z/H-ratio kernels, while the Z/H ratio sensitivity was only concentrated near the station. The reason for this is that the amplitude-sensitivity kernels of the Z-component and the H-component had the same pattern on the source side and a common propagation path, and the use of the component-differential calculation could remove the observational errors in the source and propagation path. This feature was consistent with previous

studies of the Z/H ratio, especially for 1D cases; that is, the Z/H ratio was only sensitive to the structures beneath the station.

To compare the sensitivity to the S-wave velocity in depth for these two data sets, we reduced the spatial dimension of the phase and the Z/H-ratio kernels shown in Figures 1g–j and 2g–j from 2D to 1D by laterally summing the values on the horizontal plane in each vertical grid and plotting them as a function of depth. Figure 3 shows the reduced 1D kernels at periods of 10 s, 20 s, 30 s, and 40 s for the phase (Figure 3a) and Z/H ratio (Figure 3b). The results showed that their sensitivity kernels were different but complementary. The Rayleigh wave phase was more sensitive to the deeper structure than the Z/H ratio at the same period, while the Z/H ratio had a strong sensitivity to the shallow structure (e.g., the sedimentary layers). Moreover, the phase-sensitivity kernel was much broader than the Z/H-ratio kernel for the same period, which meant the phase information could be used to solve the smoothed velocity (background model), while the Z/H ratio could deal with the lateral heterogeneity (e.g., velocity anomaly or undulated interface) better.

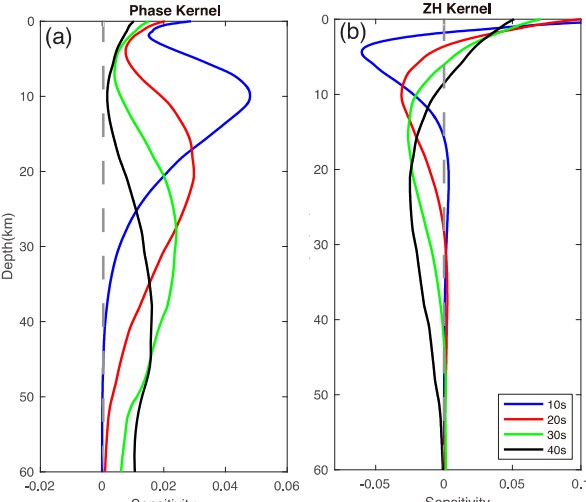

**Figure 3.** The laterally summed 1D sensitivity kernels to S-wave velocity in depth for (**a**) phase velocity and (**b**) Z/H ratio at four different periods: 10 s (blue line), 20 s (red line), 30 s (green line), and 40 s (black line).

The effect of the velocity anomaly on the Z/H ratio was obvious and easily reachable from Figure 2. Here, we designed another experiment to demonstrate the effect of undulated interface on the Z/H-ratio kernels as shown in Figure 4. We chose a typical structure containing interface variation, which followed a co-sinusoidal function with a maximum amplitude of 5.0 km. The results showed that the Z/H-ratio sensitivity was significantly affected by the undulated interface, and the maximum difference between the undulated interface and the flat interface was about 50%. The experiments described above suggested that the Z/H-ratio kernel was much more sensitive to the model with velocity anomalies and interface variations. This meant that the Z/H ratio could help us to constrain a shallow crust with a complex lateral heterogeneity in which the surface-wave phase cannot be resolved, as well as to improve the accuracy of the inversion.

**The experiment for the effect of Lateral Heterogeneity on ZH ratio kernels**

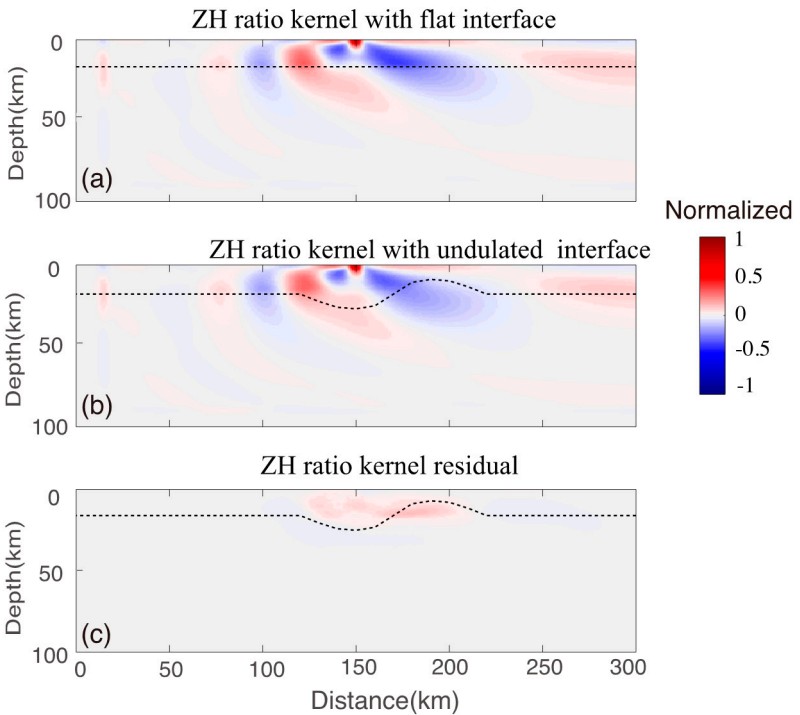

**Figure 4.** The effect of the local heterogeneities on the Z/H-ratio kernel: (**a**) Z/H-ratio kernel in the model containing a flat interface (dashed line); (**b**) Z/H-ratio kernel in the model containing an interface variation (dashed curve), which follows a cosine function with a maximum amplitude of 5.0 km; (**c**) differences between the Z/H-ratio kernels in (**a**,**b**).

## 3. Joint Inversion of the Rayleigh Wave Phase and the Z/H Ratio

As was demonstrated by the sensitivity analysis described in the previous section, the phase and the Z/H ratio of a Rayleigh wave had different sensitivities to the S-wave velocity structure, and they provided complementary constraints on the structure. Thus, joining the two data sets will help reduce the non-uniqueness of the inversion.

For adjoint tomography of the Rayleigh wave phase [32], we sought to minimize the travel-time misfits between the observed waveform and the synthetic waveform generated using SPECFEM2D. In this study, the frequency-dependent travel-time misfits can be written as:

$$\chi_{RP}(m) = \frac{1}{N_c} \sum_{i=1}^{N_c} \frac{1}{M_i} \sum_{j=1}^{M_i} \int \left( \frac{\Delta T_j(\omega, m)}{\sigma_{T,j}(\omega)} \right)^2 d\omega \, , \tag{14}$$

where $m$ is the model vector, $\chi_{RP}$ is the misfit function for the Rayleigh wave phase, $N_c$ is the number of periods, $M_i$ is the total number of misfit measurements in the $i$-th period, $\Delta T_j(\omega)$ is the frequency-dependent phase delay between the $j$th pair of observed data and the synthetic data at frequency $\omega$ in the ith period, and $\sigma_{T,j}(\omega)$ is the corresponding measurement uncertainty. Similarity, the frequency-dependent Z/H ratio misfit is defined as follows:

$$\chi_{ZH}(m) = \frac{1}{N_c} \sum_{i=1}^{N_c} \frac{1}{M_i} \sum_{j=1}^{M_i} \int \left( \frac{\Delta zh_j(\omega)}{\sigma_{zh,j}(\omega)} \right)^2 d\omega \, , \tag{15}$$

where $\chi_{ZH}$ is the misfit function for the Rayleigh wave's Z/H ratio, and $\Delta zh_j(\omega)$ is the Z/H ratio variation measured between the jth pair of the observed data and synthetic data with an uncertainty of $\sigma_{zh,j}$.

Thus, the objective function of the joint inversion based on the adjoint-state method is defined as follows:

$$\chi(m) = \omega_{RP}\,\chi_{RP}(m) + \omega_{ZH}\,\chi_{ZH}(m)\,, \tag{16}$$

where $\omega_{RP}$ and $\omega_{ZH}$ are the weights used to balance the two data types to prevent the results from being dominated by either one. The joint-inversion scheme is described in Figure 5.

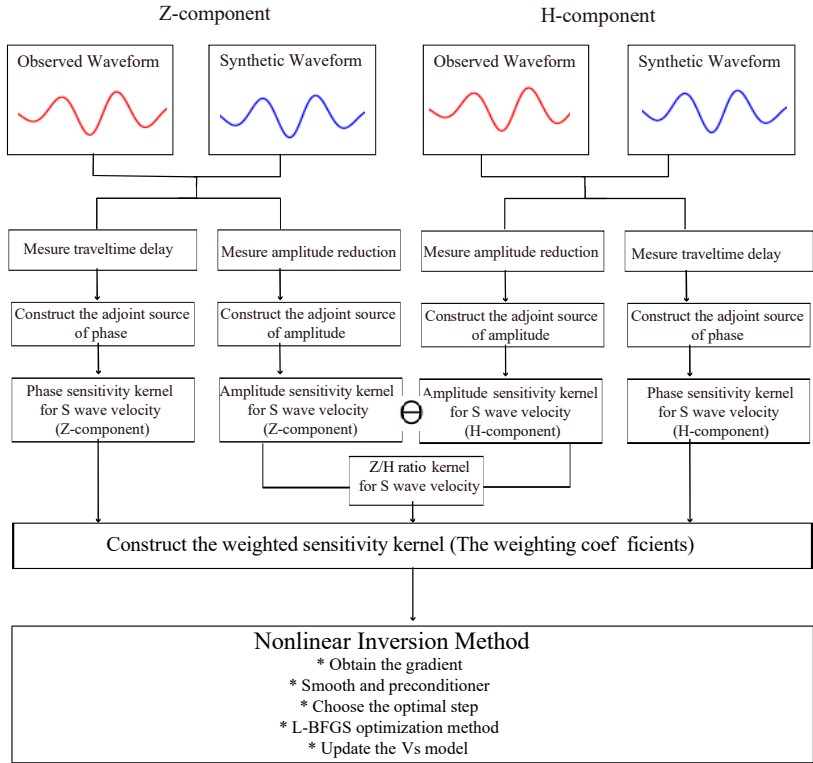

**Figure 5.** Flow chart of the 2D full-waveform joint inversion of the Rayleigh wave phase and the Z/H ratio.

## 4. Inversion Experiments

The complementary sensitivity of the phase and Z/H ratio to the vs. perturbation makes the joint inversion an important method of constraining S-wave velocity structures. Here, we designed two synthetic experiments to demonstrate the successful usage of this method on the large- and small-scale inversions.

### 4.1. Large-Scale Inversion Using Ambient Noise/Earthquake Surface Waves

Seismic linear arrays with dense station spacing have been deployed around the globe, which makes it possible to apply joint inversion to ambient-noise data or earthquake surface waves recorded by linear seismic arrays.

We set up a target model (Figure 6a) with alternating 6% box-shaped high- and low-Vs anomalies in the background model (also used as the initial model in this experiment; see material properties in Table 1), and no Vp or density perturbation was introduced into the target model. The size of the model and the number of uniform mesh nodes were the same as the parameter settings described in Section 2.3. All of the boxes with anomalies had lengths of 40 km, while the widths changed from 8 km in the upper row to 15 km in the middle row to 30 km in the lower row. These different sizes were designed to assess the spatial resolutions of the varying lengths for a Rayleigh wave with different periods. The source-time function was Gaussian, and the dominant frequency of the source was

0.2 Hz. Fifty stations were spaced 10 km apart and were located from 150 to 650 km, which could also be considered as a virtual source generating an empirical Green's function of the surface wave.

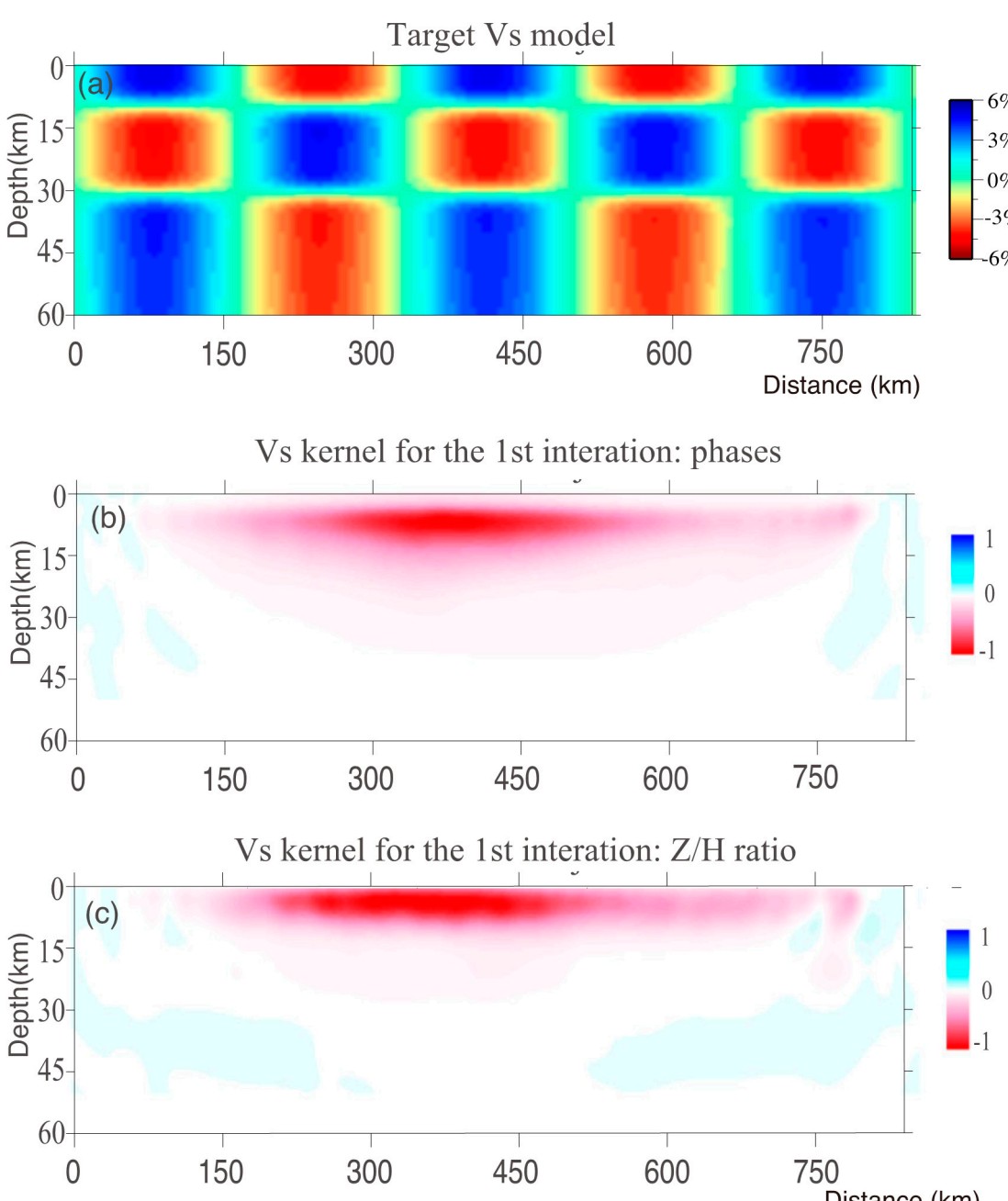

**Figure 6.** (**a**) Target model with alternating 6% box-shaped high- and low-Vs anomalies in the background model. The vs. sensitivity kernels for the first iteration of the (**b**) phase and (**c**) Z/H ratio.

We performed all of the forward and adjoint simulations using SPECFEM2D. Then, we filtered the synthetic and observed waveforms at four central period bands (10, 20, 30, and 40 s) using a Gaussian filter, which could cover the period band of 5 to 50 s quite well. Following the workflow of the joint inversion described in Figure 5, we measured the frequency-dependent travel-time and amplitude misfit between the synthetic and observed waveform and obtained the corresponding adjoint sources of the phase and amplitude, respectively. After that, we calculated the phase-sensitivity kernel and the amplitude kernel of the Z-component and H-component by injecting the adjoint sources at the stations based on the

adjoint-state method. In addition, the Z/H-ratio kernels were calculated by subtracting the Z-component and H-component amplitude kernels. Finally, all of the event kernels were summed, smoothed, and preconditioned to obtain the final misfit gradient for updating the model. A 2D Gaussian function was used to smooth the summed kernel as a regularization procedure (Tape et al., 2010). The horizontal and vertical widths of the Gaussian function used to smooth the gradient were chosen as 30 km and 10 km, respectively, in the first 10 iterations. Then, they were reduced to smaller values of 10 km and 5 km, reflecting the incorporation of the shorter-period surface-wave measurements. Next, the smoothed kernel was preconditioned using a preconditioner, which was numerically calculated through the interaction between the forward and adjoint acceleration fields; namely, the pseudo-Hessian (Luo, 2012).

$$P(x) = H_{\rho\rho}^2(x, \ x) = \sum_{s=1}^{N_s} \int \partial_t^2 s(x,t) \cdot \partial_t^2 s^{\dagger}(x, T-t) dt \ , \tag{17}$$

where $s$ and $s^{\dagger}$ are the forward and adjoint displacements, respectively, and $N_s$ is the number of sources. Finally, we used the limited-memory Broyden–Fletcher–Goldfarb–Shanno (L-BFGS) method [43] (see detail shown in Appendix A) to update the model, and a line-search method was used to determine the optimal step length for updating the model. Once the optimal step length was determined from the minimum of the total function for all of the bands, the model parameter m was updated as follows:

$$ln \ (m_{i+1}/m_i) = v_i \ d_i \ , \tag{18}$$

where $m_{i+1}$ and $m_i$ are the models of the (i + 1)th iteration and the $i$th iteration, respectively; and $v_i$ and di are the optimal step length and the search direction (i.e., summed event kernels) of the $i$th iteration, respectively.

To demonstrate the advantage of our joint-inversion framework, we also conducted two additional separate inversions using either only the Rayleigh-wave phase data or only the Z/H-ratio data. The separate inversions also began with the background model and used the same inversion parameters as the joint inversion, including the smoothing width and step lengths. Both data sets were mostly sensitive to the shear-wave velocity. We only inverted for the vs. structures. In total, we conducted three inversions: (1) adjoint tomography of the Rayleigh-wave phase data (i.e., phase inversion); (2) adjoint tomography of the Rayleigh-wave Z/H-ratio data (i.e., Z/H-ratio inversion); and (3) the joint inversion using the two data sets (i.e., joint inversion).

Figure 6b,c show the gradients of the vs. for the first iteration of the phase and Z/H-ratio inversions. The gradients show that the two Rayleigh wave measurements had different and complimentary sensitivities to the crustal structure. The phase inversion was more sensitive to the deeper structure than the Z/H-ratio inversion, while the Z/H-ratio inversion had a strong sensitivity to the shallow structure. The total misfit evolution of the phase and Z/H-ratio inversions and the joint inversion are presented in Figure 7a,b. We can see that the joint inversion had a slower convergence rate and slightly larger misfits than those of the separate inversions, which was reasonable, since the joint-inversion scheme tried to fit both data sets simultaneously. The joint inversion converged after 32 iterations when the misfit changed over the last iteration because both the phase and Z/H-ratio data were less than 3%. Figure 7c,d show the waveform fitting of the observed Z-component and X-component data and the two synthetic computed data sets based on the initial and final models for the source–receiver pairs located at (750 km, 0 km) and (150 km, 0 km), and significant improvements in the travel-time and amplitude fits after joint inversion can be observed.

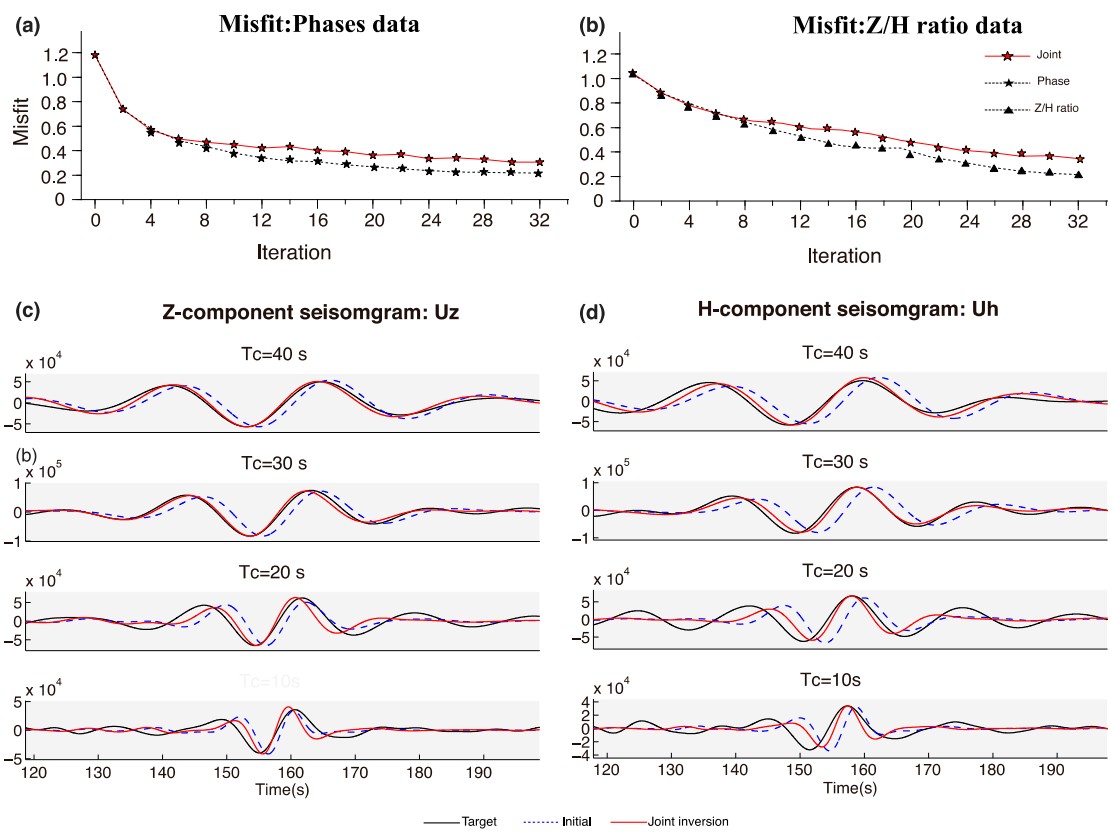

**Figure 7.** The total misfit evolution for the (**a**) phase and (**b**) Z/H ratio along with iterations in the phase inversion (black stars), the Z/H-ratio inversion (black triangles), and the joint inversion (red stars). The seismic traces obtained using joint inversion (red solid line) and the source–receiver pairs at (750 km, 0 km) and (150 km, 0 km) are shown together with the seismic traces of the initial (blue dashed line) and target (black solid line) models for the (**c**) Z-component and (**d**) H-component.

To compare the results of the three types of inversions, we show their final vs. models in Figure 8. The Z/H-ratio information for the surface was mostly sensitive to the vs. structures at shallow depths (<30 km), which was limited by the period range of the Rayleigh wave in this study. Thus, the recovered vs. model (Figure 8b) was better resolved in the shallow regions, and the upper and middle rows show the correct pattern with the anomaly close to target model. Compared with the Z/H-ratio inversion, the phase inversion was sensitive to much deeper vs. structures and could constrain the broader range of the vs. structures at depths of ~20 km to 60 km (Figure 8a). The pattern of the anomaly in the middle and lower rows is almost recovered; however, the recovered pattern is rather smoothed and lacks constraints in the shallow structure, which was caused by the sensitivity characteristic of the phases shown in Figure 3a. Benefiting from the sensitivity to the vs. structures concentrated near the station, the Z/H-ratio inversion helped to better resolve the structures in the shallow areas that were not well resolved by the phase inversion. Thus, the addition of the Z/H ratio in the joint inversion helped to constrain the crustal structure at shallow depths. At greater depths, the joint inversion had a better resolution than the phase inversion with the correct amplitude recovery. These tests demonstrated that the joint inversion combined the complementary sensitivities of the surface wave phases and Z/H ratio, and was capable of building a more-unified vs. model, thus outperforming the inversions based on the individual data sets.

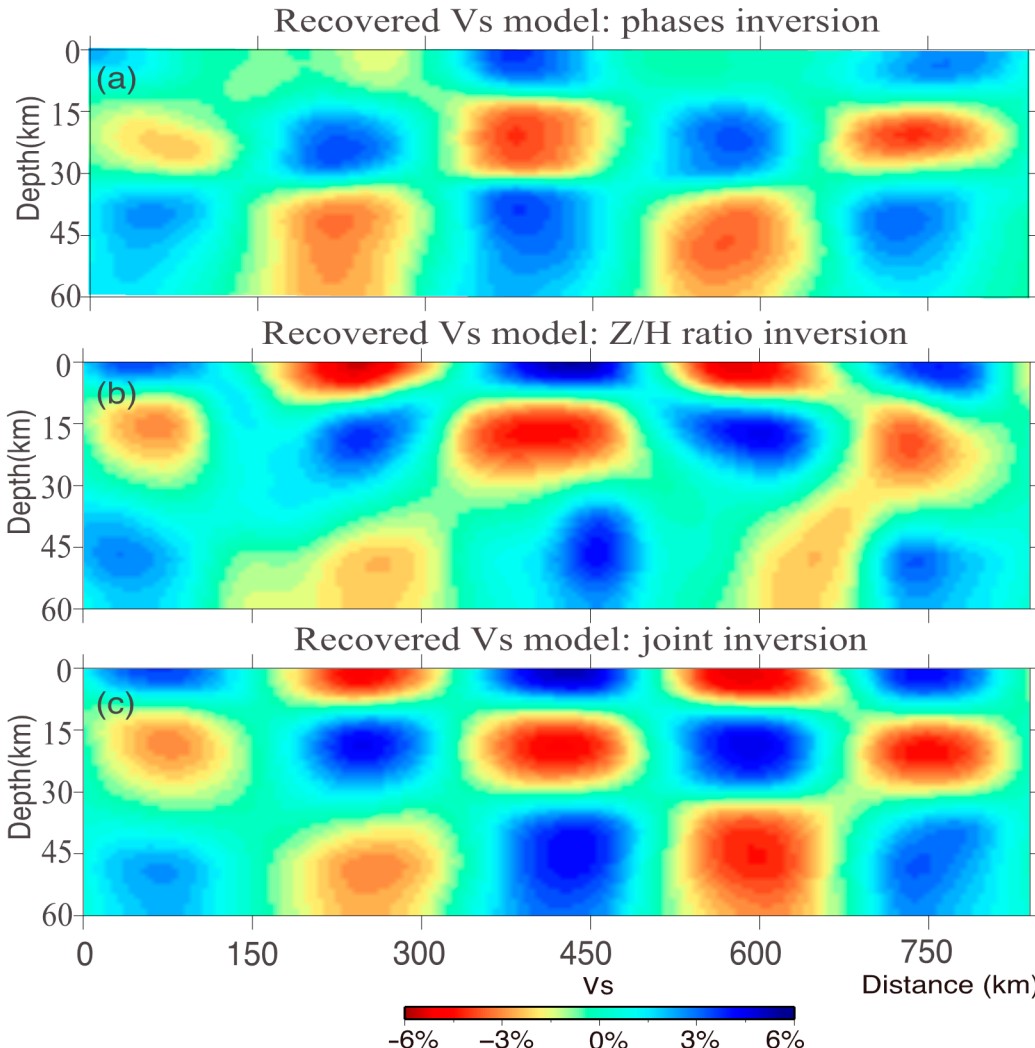

**Figure 8.** The recovered S-wave velocity models for the three types of inversions. (**a**) The final inverted model using only the Rayleigh wave phase. (**b**) The final inverted model using only the Rayleigh wave Z/H ratio. (**c**) The final inverted model using the joint inversion of the Rayleigh wave phase and Z/H ratio.

*4.2. Small-Scale Inversion Using Active Sources*

Another suitable application of this method is retrieving model parameters and characterizing the near-surface region, mainly the shear-wave velocity model using active sources, for engineering applications, geotechnical studies, and physical modeling tests. In this section, we used more realistic synthetic experiments to demonstrate the promising applications of this full-waveform joint-inversion method to small-scale inversions, with an improved resolution.

The $600 \times 100$ m target model for elastic S-wave and P-wave velocities used for the numerical experiments in this section is shown in Figure 9. With this model, we used 100 and 20 uniform mesh nodes in the horizontal and vertical directions, respectively, resulting in a total of $(4 * 100 + 1) * (4 * 20 + 1) = 32{,}481$ unique grid points. The target elastic model included heterogeneous Vp (Figure 9a) and vs. models (Figure 9b) with a homogeneous density of 1000 kg/m$^3$. Both the Vp and vs. increased with depth from a low-velocity layer, with the exception of one low-velocity zone (LVZ) and two high-velocity zones (HVZs) in the middle. We assumed an initial model with a homogeneous Vp of 1000 m/s, a homogeneous vs. of 500 m/s, and a homogeneous density of 1000 kg/m$^3$. We set up 50 shots located from 50 to 550 m at a depth 0.5 m below the model's surface, with an equal

10 m spacing. The stations were spaced 5 m apart and were located 5 m to 595 m at a depth of 0.5 m.

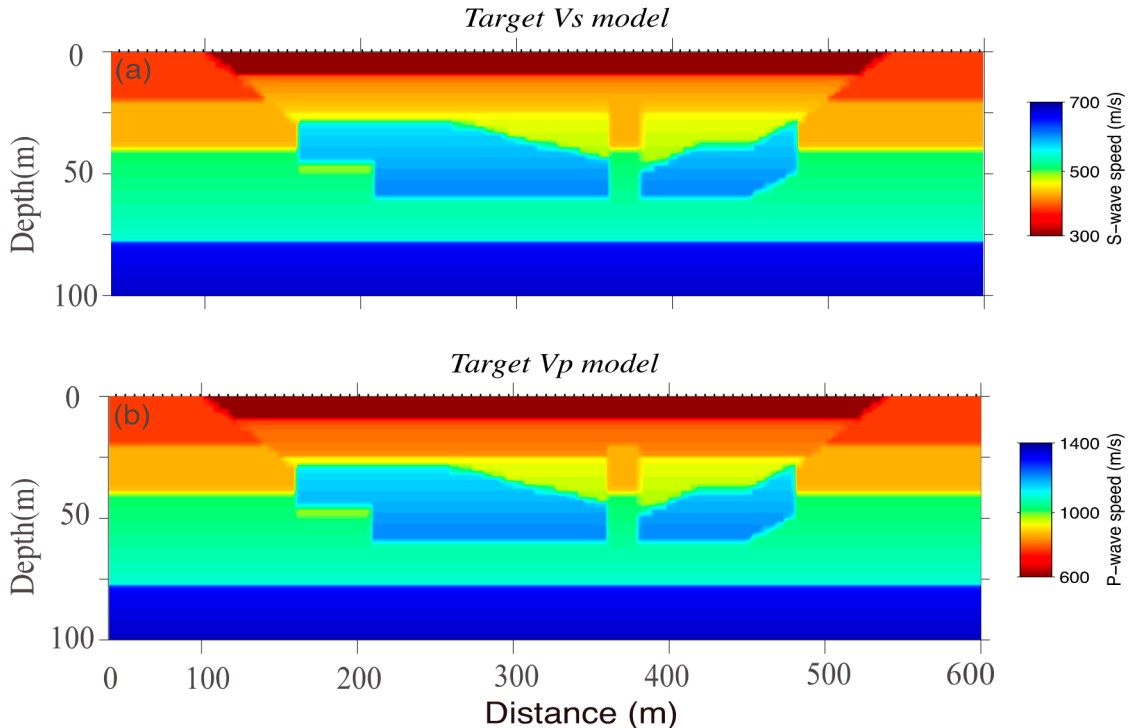

**Figure 9.** Target model with features in the near surface, specifying (**a**) S-wave and (**b**) P-wave velocities.

Synthetic data were computed for the target model and the initial model and were simulated using source-time functions with a dominant 10 Hz Ricker wavelet. Then, we conducted the joint inversion following the same inversion procedures described in Figure 5. To avoid the local minima and reduce the non-linearity of the waveform misfit function, we adopted a hierarchical strategy, in which we successively inverted from low to high frequency progressively through 20, 30, 40, 50, 60, and 80 Hz. We also conducted three types of inversions to further demonstrate the advantage of the joint inversion over the separate inversions. For each inversion, we simultaneously inverted both the S-wave and P-wave velocities.

First, we used the objective function in Equation (13) to update the model via the phases. The result of using the Rayleigh wave phase is shown in the inverted S-wave (Figure 10a) and P-wave (Figure 11a) velocity models. A better result for the vs. final model was expected, since the surface waves were more sensitive to this elastic parameter. Neither the S- nor the P-wave velocity model converged to a model near the target shown in Figure 9. The most probable reason for this was that the inversion suffered from strong cycle-skipping problems, which easily occur in surface waveform FWI, with incorrect updates leading to a secondary minimum. Next, we replaced the phases with the Z/H ratio in the objective function. That is, we minimized the discrepancy defined by the Z/H ratio in Equation (14). Figures 10b and 11b show the resulting vs. and Vp models. Because the envelope information using the Z/H ratio calculation in this paper mitigated the cycle-skipping effects, working only with the Z/H ratio led to a reasonably successful recovery of the vs. model of the shallow structure, but it did not constrain the deeper structure. Moreover, the capacity of the Z/H ratio to constrain the shallow crust with a complex lateral heterogeneity helped us obtain a significantly better low-velocity layer where the surface wave phase could not resolve the structure. The S-wave velocity structure was resolved, albeit with a low resolution, and only the very top portion of the P-wave velocity model showed signs of being updated in the desirable direction. The Z/H ratio inversions

appeared to converge at the price of losing resolution by discarding the phase information. Finally, we proposed a full-waveform joint inversion of the phase and Z/H ratio in the same multiscale framework. Figures 10c and 11c show the results obtained using the joint-inversion philosophy. The addition of the Z/H ratio to the joint inversion helped constrain the S-wave velocity structure in the near-surface region, so the low-velocity layer was correctly corrected and converged to the target model. Moreover, the mitigation of the cycle-skipping problems contributed to the Z/H ratio and could help the phase information better constrain the principal features of the deeper structure. The LVZ and HVZ in the middle were well covered with correct amplitudes. In addition, the interfaces between the velocity zones were better defined in the vs. model. In the Vp model, although only the very top portion of the P-wave velocity model was acceptably imaged, it can be seen that the inclusion of the phase information in the joint inversion provided more additional structure to the P-wave velocity model compared with the inversion results obtained using on obtained using only the Z/H ratio.

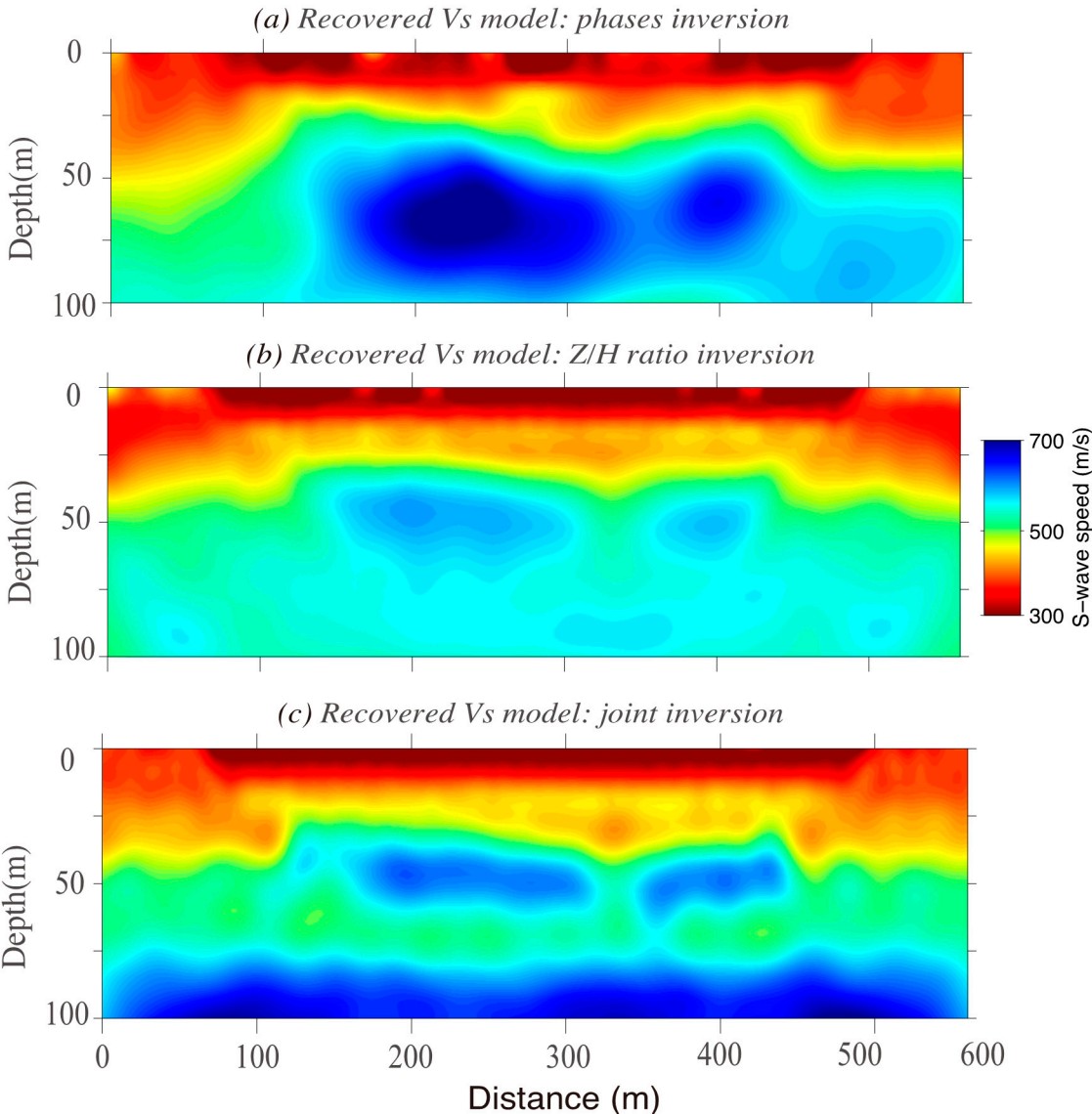

**Figure 10.** The recovered S-wave velocity models for the three types of inversion. (**a**) The final inverted model using only the Rayleigh wave phase. (**b**) The final inverted model using only the Rayleigh wave Z/H ratio. (**c**) The final inverted model using the joint inversion of the Rayleigh wave phase and Z/H ratio.

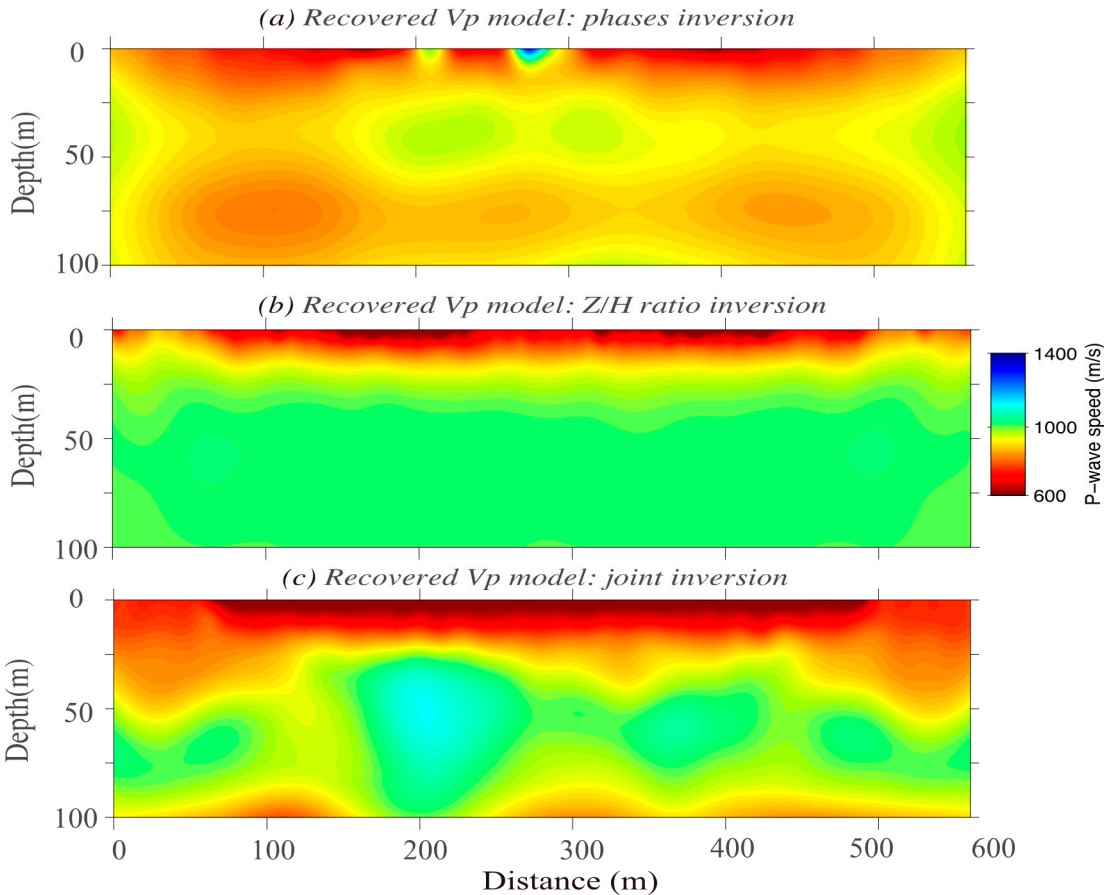

**Figure 11.** The recovered P-wave velocity models for the three types of inversions. (**a**) The final inverted model using only the Rayleigh wave phase. (**b**) The final inverted model using only the Rayleigh wave Z/H ratio. (**c**) The final inverted model using the joint inversion of the Rayleigh wave phase and Z/H ratio.

In Figure 12, we show how well the waveform data were fit under phase and joint inversion as described in this paper. The left column shows the vertical displacement components (Uz), and the right column shows the horizontal components (Uh) for three traces of stations at X = 150, 300, and 450 m from one shot gather located at X = 50 m. The black solid lines represent the target data, red dashed lines represent the initial predictions, and blue solid lines and green solid lines represent the final models obtained by phase inversion and joint inversion, respectively. We can see that the phase inversion suffered from a cycle-skipping problem, creating local minima in the FWI objective function, so the final traces were badly matched to the target traces (Figure 12a). Joint inversion incorporating Z/H ratio improved the vs. model by mitigating cycle-skipping issues, so the traces of joint inversion were matched to almost-perfect agreement with the target traces (Figure 12b).

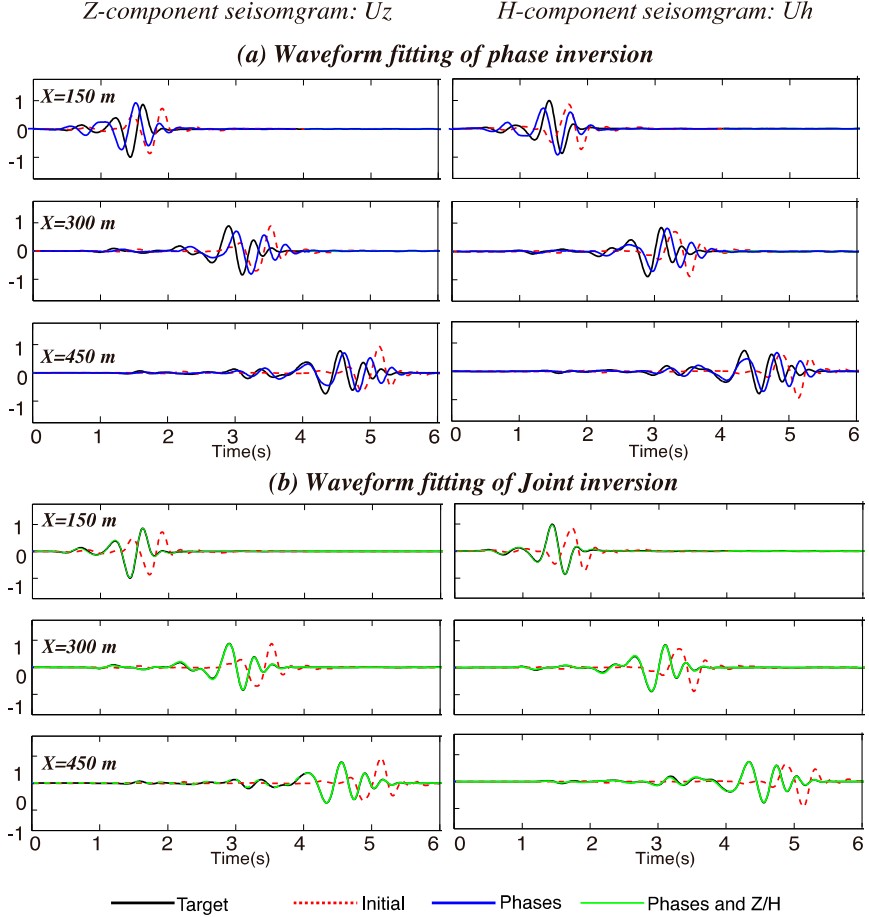

**Figure 12.** Waveform fitting for three traces of stations at X = 150, 300, and 450 m from one shot gather located at X = 50 m after inversion. Comparison of the target, initial, and final seismic traces obtained via (**a**) phase inversion; (**b**) joint inversion. Left: H-component. Right: Z-component.

## 5. Discussion

As the extension of 2D surface-wave adjoint tomography method [32], the computing of 2D joint inversion also kept the efficiency. Taking the small-scale inversion as an example, the model was 600 m in length and 100 m in depth with $100 \times 20$ elements, and the simulation contained 20,000 time steps with a 0.0003 s time interval. As shown in Table 2, the phase inversion included 50 forward-modelings and calculations of kernels; the number of processors for each source was 4, leading to a total cost of 200 CPU hours. The computation time was 6.25 h for 46 iterations. For joint inversion, the CPU hours cost was doubled (400 CPU hours total) due to the addition of the Z/H-ratio kernel calculation. However, the actual computation also showed relative efficiency, and 38 iterations could be completed in just 5.8 h, which was acceptable compared with the computational cost in the 3D case. On the basis of the time cost, the 2D joint inversion can be very efficient when computational resources are limited.

**Table 2.** Parameters of numerical modeling and computational time for inversion.

| Type of Inversion | Phase Inversion | Z/H Ratio Inversion | Joint Inversion |
|---|---|---|---|
| Model size (Nx ∗ Nz) | x-direction: 600 m; z-direction: 100 m Nx = 100; Nz = 20 | | |
| Propagation time | 20,000 time steps; Time discretization: 0.0003 s | | |
| Number of processors for each source | 4 | 4 | 8 |
| Total number of CPUs/cores | 200 | 200 | 400 |
| Number of iterations | 46 | 54 | 38 |
| Time for inversion | 6.25 h | 6.72 h | 5.80 h |

The synthetic examples above provided us with two actual situations for the application of joint inversion. For large-scale inversion, we could use earthquake surface waves recorded by linear seismic arrays. In reality, it is difficult to find enough earthquakes along the profile. Fortunately, the approach has been developed over decades, in which the Z/H ratio can be extracted from ambient-noise cross-correlation data [7], which makes it the most suitable situation for applying the joint-inversion method. The tomographic experiment in Section 4.1 shows that both Rayleigh wave phase and Z/H ratio had sensitivity to S-wave velocity in the crustal structure, but their sensitivity patterns were different. Rayleigh wave phase velocity dispersion was sensitive to the average velocity of the depth range sampled. The Z/H ratio was more sensitive to the shallower structure compared to the Rayleigh wave phase velocity dispersion at the same frequency; thus, the Z/H ratio was more helpful to constrain the shallow structure, such as the sedimentary basin. The joint inversion combined the complementary sensitivities of the phases and Z/H ratio and was capable of building a more unified crustal structure.

For small-scale inversion, this method could be used effectively in retrieving model parameters and characterizing the near-surface region using active-source surface waves for engineering applications, geotechnical studies, and physical modeling tests. The tomographic experiment in Section 4.2 indicated that joint inversion has another important advantage; that is, it can deal with the cycle-skipping problem, which easily occurs in FWI of surface waves in near-surface cases. This contributes to envelope-information usage in the Z/H ratio calculation, which avoids incorrect vs. model updates, leading to a secondary minimum. Due to limited sensitivity to the P-wave velocity model, it was found that the P-wave model also seemed to be not well constrained by the joint inversion. This can be improved by inclusion of body-wave data sets, such as the receiver function for large-scale inversions and seismic refractions for small-scale inversions, which is sensitive to the P-wave velocity.

Before application to real data, there are some issues that require more study. One is the difficulty of extracting stable Z/H-ratio measurements from a single source–station pair. Similarly, using amplitude information from an ambient-noise cross-correlation is not always reliable. Another issue is the 3D structural effect on both phase and Z/H-ratio kernels, which were not properly modeled in the 2D case. For example, the ray path may not follow the profile. Thus, the requirements for the linear profile need to be considered before a field experiment is conducted. In addition, the real data FWI is indeed more difficult in near-surface application because of many practical problems, such as waveform data-signal processing problems in signal-to-noise ratio improvement, the effect of rugged topography beneath real irregular receiver array acquisition, a very complicated near-surface effect, the random or man-made noise effect, and heterogeneous distribution of ambient-noise sources. Ignoring any issue in the real data waveform inversion could cause the inversion result to be unexplained or unstable. This paper focused on the theoretical

and technical aspects, and the real data application will be discussed and published in future works.

## 6. Conclusions

In this study, we developed a full-waveform joint-inversion scheme for the Rayleigh wave phase and the Z/H ratio simultaneously. First, we developed an approach for calculating the 2D sensitivity kernels of the Rayleigh wave phases and the Z/H ratio based on the adjoint-state method. A numerical experiment was carried out to investigate the sensitivities of the Rayleigh wave phase and the Z/H ratio to the S-wave velocity structure. The experiments showed that they were both mostly sensitive to the shear-wave velocity, but their sensitivity patterns were different. The phase kernel was concentrated along the propagation path, with a much broader sensitivity range than the Z/H-ratio kernels. The Z/H ratio was more sensitive to the shallower structure compared to the Rayleigh wave phase-velocity dispersion at the same frequency. In addition, the Z/H ratio could deal with the lateral heterogeneity better, which was more helpful in constraining shallow structures, such as sedimentary basins or near-surface structures.

The joint-inversion method was applied in two synthetic experiments involving large- and small-scale inversions. By comparing the tomographic results of the adjoint tomography of the Rayleigh wave phases, the adjoint tomography of the Z/H ratio, and the joint inversion, we demonstrated that the joint inversion outperformed the separate inversions because it combined the complementary sensitivities of both, resulting in a more S-wave velocity unified model. Also, joint inversion could deal with the cycle-skipping problem in FWI in near-surface cases. The proposed joint-inversion scheme offers a computationally efficient and inexpensive alternative to imaging fine-scale crustal structures or near-surface structures beneath a 2D seismic array.

**Author Contributions:** C.Z. made the main contributions to this research, providing the ideas of modeling and inversion; T.L. was responsible for the simulation; Y.W. provided suggestions on the feasibility and innovation of the solution. All authors have read and agreed to the published version of the manuscript.

**Funding:** This research was funded by financially supported by the Natural Science Foundation of Jiangsu Province, China, grant number BK20190499; the National Natural Science Foundation of China, grant number 42004037; and the Fundamental Research Funds for the Central Universities grant number 2019B0071428.

**Institutional Review Board Statement:** Not applicable.

**Informed Consent Statement:** Not applicable.

**Data Availability Statement:** The data of this study is available from the authors upon request.

**Acknowledgments:** We gratefully acknowledge Qinya Liu's constructive comments from university of Toronto.

**Conflicts of Interest:** The authors declare no conflict of interest.

## Appendix A

*Limited-Memory BFGS Algorithm*

The main algorithm is: given an initial model $m_0$, objective function f, diagonal scaling D, memory value l, and stopping threshold $\delta > 0$, the L-BFGS algorithm is as follows:

(1)  Evaluate $f_0 = f(m_0), g_0 = \nabla f(m_0)$;
(2)  Set $p_0 = -g_0, k = 0$;
(3)  If k > 0, compute $p_k$ from recursion, below;
(4)  Compute $\alpha$k by line search and set $m_{k+1} = m_k + \alpha_k p_k$;
(5)  Evaluate $f_{k+1} = f(m_{k+1}), g_{k+1} = \nabla f(m_{k+1})$;
(6)  Set $s_k = m_{k+1} - m_k, y_k = g_{k+1} - g_k, k = k + 1$;
(7)  Repeat (3–6), stopping immediately when $g_{k+1}^T g_{k+1} < \delta$;

Recursion

(1) Set $q = g_k, i = k - 1, j = min(k, l)$;

(2) Perform j times: $\lambda_i = \frac{s_i^T q}{y_i^T s_i}, q = q - \lambda_i y_i, i = i - 1$;

(3) Set $\gamma = \frac{s_{k-1}^T y_{k-1}}{y_{k-1}^T y_{k-1}}, r = \gamma D q, i = k - j$;

(4) Perform j times: $\mu = \frac{y_i^T r}{y_i^T s_i}, r = r + s_i (\lambda_i - \mu), i = i + 1$;

(5) End with result $p_k = -r$;

L-BFGS's relatively modest memory usage rests on the fact that if k is the current iteration number and l is the memory value, then vector pairs prior to {sk−l,yk−l} are no longer needed and can be removed from storage. The scaling factor γ, which accounts for differences between the true Hessian and the approximation thereto, is essential to the good performance of the algorithm.

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
