# Peer review of "Two-Dimensional Full-Waveform Joint Inversion of Surface Waves Using Phases and Z/H Ratios"

_applsci, doi:10.3390/app11156712_

Round 1

Reviewer 1 Report

The paper is well articulated. The introduction is clear, the conclusions are well structured. The theory is presented in a very satisfactory way. The graphics are clear

Author Response

Thanks for your positive comments and great suggestions on our manuscript paper.   We have carefully check the English language and style in this paper and revised the corresponding part.

Reviewer 2 Report

In general the paper is well written and it deserves publication. 

However, few amended are necessary in order to improve further the manuscript. 

Minor suggestions:

Figure 4: Include labels on axis

Figure 5: Maybe font size can be increased  making the diagram more readable 

Author Response

Minor suggestions:

Figure 4: Include labels on axis

Reply: Thanks for your advices.

We have already revised the labels on axis in Figure4.

Figure 5: Maybe font size can be increased  making the diagram more readable

Reply: Thanks for your advices.

We have already increased font size in table1, the corresponding revisions can be found in the main text at line 239.

Reviewer 3 Report

Review of the manuscript Two-dimensional full-waveform joint inversion of Surface waves using phases and Z/H ratios by Chao Zhang, Ting Lei, Qinya Liu, Yi wang

I am a physicist with strong experience in crustal structure imaging with various seismic methods.

General impresion

This manuscript describe comparison of three types of inversion using phase of Rayleigh waves, Z/H amplitude ratio of it and the combination of both in join intersion. The idea is clear and as sensitivity kernels of both components are different the joint inversion will lead to better results. Article shows that using two synthetic examples for large crustal model and small near-surface model. In my opinion figures 1 and 2 explains sensitivity kernels and figures 8, 10 and 11 proves that the metod is working in two scales. The other figures and most of the text are not needed to pass this message. The manuscript deliver one more important result, that proposed joing method can deal with the cicle-skipping problem in FWI in near-surface cases. Unfortunately this result was not mentioned in conclusions.

The manuscript is written in very good English (please note I am not a native speaker, but I have no problem understanding it). All figures are great quality and prepared according to best publication standards.

Title and abstract are informative, and the introduction is an excellent explanation of development and the state of the art methods with well selected references. Unfortunatelly the Discussion is almost not existing and the conclusion is limited to three sentences.

Major problems

  • The article is too long. Many aspects is presented using multiple figures that are not necessary:

Figure 7 shows that using two datasets with different senditivity kernel  gives better results that one data set, repeating the main message of the article.

Figure 12 is not needed. The same conclusions can be reached from Figures 10 and 11. It is easy to sea that sharp boundaries are not recovered.

Similarly Figure 13 shows nicely the waveform difference for different approaches. But why it is presented here? Is this article showing the joint method that better reconstructs teh structure or how obtained structural models reconstruct synthetic wavefield. All of it looks great, but it is long and I dont see any clear mesage in it.

Figure 14 also nicely shows the proces, but gives nothing to the whole message.

  • All synthetic calculations are based on very good and widely used SPECFEM2D software, and all methods were previously published and are precisely refered in chapters 2 and 3. Also joint interpretation of phase and Z/H ratio is not new. It is difficult to me to understand what is new in this article and what new value authors would like to present. This should be clearly demonstrated. Application of existing method to two simple synthetic tests to show effects that are obvious from the method itself is not good enough.

  • I dont see sense in equations 6 – 11, because eq 12 explain what is in this paper (simple envelope ratio). Maybe I did not understand this message and it should be better explained.

Minor sugestions:

I dont understand the reason to show laterally summed 2D data in 1D plot in Figure 3. Full 2D results presented in Figure 2 are much more informative and the same conclusions can be reached. I suggest to remove Figure 3.

Figure 4 shows 2D sensitivity to anomalies (synthetic test). This result is obvious and easily reachable from figures 1 and 2

Line 35 is ambigous – „due to its lower frequency” suggests that the crust have frequency, i suggest to change it to „due to low frequency of such waves”.

Line 286 – specfem2D should be written with capital letters

Line 317- mode -> model

Round 2

Reviewer 3 Report

In a previous review a few days ago I recommended rejecting this manuscript pointing out several problems. I think that the manuscript in its current form is very well written and beautifully illustrated. It has an excellent introduction as well. The problem is that it has no scientific content authorizing the publication of the results. The main message of this article is „using two data set with different sensitivity kernel gives better results than using only one data set” and I fully agree with that, but you don't need a 23-page manuscript and 12 Figures to show that. This one sentence is sufficient. I think that changing this manuscript into a short note (few pages, and figures 1, 2, 8, 10 and 11) is out of the scope of major revision so I recommended rejecting and resubmit it.

I expect the Applied Sciences journal (Q2, IF=2.5) should provide readers with interesting scientific content, so again I will point out problems in this manuscript:

  1. Synthetic data only

It is a simple synthetic test. There are no experiments as mentioned in the text. To prove it you need to apply it to real data.

  1. both cases presented in the manuscript are perfect synthetic data tests

I know that SPECFEM is working well as it is being developed for many years by strong teams in ETH and Princeton. Showing on perfect synthetic data that it is working as intended, proves nothing. Not even noise was simulated to represent a real situation.

  1. proposed new approach (eq. 13)

If you propose a new approach you should show what is wrong with the old one (eq.12) with a simple concept of envelopes, that is widely used. What is your method with complicated integrals doing better? Is it faster, more precise, easier to implement, or does it handle poor S/N data in a better way? I don't get it from the article, and updated text (168-169) that both methods can get a similar Z/H ratio is not convincing. So what is the point of doing it?

  1. FWI benefit in near-surface

If you claim that this approach will support FWI analysis please show it on an example of FWI analysis, but not using perfect synthetic data. The real example would be the best, but that is indeed difficult in near-surface because of many practical problems.

I will not be able to help further with this article. Modifying it further will not improve the content. The form of it is very good. I refuse to review it further. If my suggestions are wrong I encourage the Editor to skip them and publish the article in its current form.
